

# Head louse egg and nit remover—a modern "Quest for the Holy Grail"

Elizabeth R. Brunton[1,2], Ian P. Whelan[2,3], Rebecca French[4], Mark N. Burgess[4] and Ian F. Burgess[1,2]

[1] Insect Research & Development Limited, Cambridge, Cambridgeshire, United Kingdom
[2] EctoMedica Limited, Cambridge, Cambridgeshire, United Kingdom
[3] Avisius Research Limited, Coventry, United Kingdom
[4] Medical Entomology Centre, Cambridge, Cambridgeshire, United Kingdom

## ABSTRACT

**Background.** The eggs of head lice are fixed to the hair of their hosts by means of a persistent glue-like fixative that is not chemically bound to the substrate. Eggshells stuck to hairs after successfully treating the infestation are a cosmetic issue and a source of misunderstanding about whether the infestation is eliminated. Hitherto, no effective treatment to loosen louse eggs and nits has been found.

**Methods.** An extensive screening of surface active compounds, oils, esters, and other cosmetic lubricants used a slip-peel device to measure the forces required to release the grip of the fixative. Promisingly effective compounds were formulated into suitable carriers for further testing. The most effective combination formulation was tested, as a commercial product (Hedrin Stubborn Egg Loosening Lotion), in a usage study of 15 children with nits, in which one half of the head was combed only on damp hair and the other half combed after a 10 min treatment using the product.

**Results.** Laboratory tests of the forces required to remove nits found that pelagonic acid derivatives, particularly isononyl isononanoate, in the presence of a polymeric gelling agent and water, were most effective to reduce the initial grip of the fixative as well as reducing friction as the eggshell is drawn along the hair shaft and that the final product was significantly ($p < 0.05$) more effective than several other marketed materials. In the usage study significantly ($p = 0.01046$) more louse eggs and nits were removed after treatment with the gel.

**Discussion.** The product developed through this study is the first with a demonstrable efficacy for loosening the grip of the louse egg fixative from hair. Consequently, until now, and despite the availability of effective pediculicidal treatments, dealing with the eggshells persisting after an infestation has been an onerous task for most households. This type of product can enable families to deal more easily with persistent eggshells and improve self-esteem in affected children.

Corresponding author
Ian F. Burgess,
ian@insectresearch.com

# INTRODUCTION

Insects of the order Phthiraptera all lay eggs that are fixed either to the hairs or the feather barbs of their hosts by means of a glue-like substance that is deposited and sets extremely

rapidly as the egg is laid (*Buxton, 1947*). Studies of this fixative material indicate that it is constructed of linear polymerized proteinaceous molecules (*Schmidt, 1939*), forming β pleated sheets (*Burkhart & Burkhart, 2005*). However, there is no evidence of a chemical bond with the hair, so that the firm positioning appears to be entirely due to a vice-like grip on the hair shaft initiated as the fixative polymerizes (*Burkhart et al., 1998*; *Burkhart et al., 1999*). The hardened glue-like material is extremely resilient, not susceptible to biological or chemical breakdown (*Burkhart et al., 1998*), and holds the empty eggshell *in situ* long after the nymph has emerged.

In the treatment of infestations of the human head louse, *Pediculus humanus capitis*, there have been several recent advances to address problems caused by acquired resistance to conventional insecticides (*Feldmeier, 2014*). Consequently, curing an infestation can be relatively easy but an age-old problem persists. After elimination of the lice, the dead eggs and empty eggshells (nits) remain firmly fixed to the hair. Numerous products have been marketed claiming to facilitate nit removal by dissolving the glue, digesting it with enzymes, or making it slide off the hair, but objective tests of these materials have shown that none of them really works any better than, or even as well as, off-the-shelf conditioning rinses or even just water (*Burgess, 2010*; *Lapeere et al., 2014*; *Ortega-Insaurralde et al., 2014*) and some products, such as those based on essential oils, may actually bind louse eggs more firmly to the hairs (*Burgess, Brunton & Burgess, 2016*). Because none of the common hair treatments exhibits any detectable influence on that grip of the egg fixative material, it has created a challenge for inventors that is not helped by the conflicting and contradictory statements and claims about the nature of the fixative (*Schmidt, 1939*; *Barat & Scaria, 1962*; *Carter, 1990*; *Burkhart et al., 1998*; *Burkhart & Burkhart, 2005*; *Federal Trade Commission, 1998*), which is why there have been many attempts to discover a chemical or best method to initiate release of louse eggshells from hair (*Greene, 1898*; *Sacker, 1942*; *Bernstein, 1990*; *Upton, 1994*; *Hayward & Watkins, 1999*; *McGuire & Kross, 2002*; *Ozelkan, Zhang & Malayev, 2003*; *Acevedo, 2010*; *Mehlhorn et al., 2013*; *Kolender & Kolender, 2017*), a complex story we likened to the Arthurian legendary "Quest for the Holy Grail". This project was initiated with the aim of identifying one or more chemicals effective to loosen the grip of the louse egg fixative in order to develop a genuinely effective consumer product.

## MATERIALS & METHODS

### Louse eggs on hair

Louse eggs and nits on hair were obtained from a laboratory culture colony of *Pediculus humanus humanus* by providing actively laying female lice with washed, untreated, human hair over a 24 h period. At the end of this time the lice were removed and the eggs and hairs frozen at −18° Celsius overnight, which kills the embryos but does not affect the resilience of the fixative (*Burkhart et al., 1998*), after which the eggs and hairs were stored in closed sealable polyethylene bags at room temperature until required. If nits were required the eggs were incubated at 30° ± 2° Celsius until all nymphs had emerged. The empty eggshells on hairs were then stored in the same way as for intact eggs.

## Laboratory testing of egg and nit removal

We used a modified SP-2000 slip-peel tester (IMASS, Inc., Accord, MA, USA) to measure the force required to pull louse eggs or nits along treated hairs as previously described (*Burgess, 2010*). In early screening tests, each hair bearing a louse egg or nit was threaded through a glass 1 μL Microcap$^{TM}$ tube (Drummond Scientific Co., Broomall, PA, USA) that had been cemented to a glass microscope slide fixed to the platen of the slip-peel tester (*Burgess, 2010*). The end of the hair at the opposite end of the tube from the louse eggshell was held in a clamp attached to the force transducer and the platen set in motion. The hairs were aligned so that the proximal end of the eggshell, the part fixed to the hair by the glue-like material, was closest to the tube. As the base of the eggshell was brought into contact with the end of the Microcap$^{TM}$ tube, force was exerted to cause the glue fixative to release after which the eggshell was able to slide along the hair shaft. This method was used for the screening of technical substances and frame formulations to ensure that any changes induced by the applied chemicals in the forces required to initiate movement of the eggshells could be detected, relative to the same forces required to remove untreated eggs. Later tests of selected formulated materials followed the same principle except the hair shaft was fitted between the close-spaced teeth of a metal comb (Innomed$^{TM}$ Lice Comb, Hogil Pharmaceutical Corp., White Plains, NY, USA), which had been fixed across the platen of the slip-peel tester (Fig. 1A). In some cases the treated eggshells, particularly the nits, could flex and slip between the teeth of the comb, unlike when the base of the egg was trapped against the end of the Microcap$^{TM}$ tube, and therefore gave a better representation of how any particular treatment formulation under test might perform if it were used with a comb to remove eggs from a head of hair.

Two forces were measured and recorded from the digital display of the slip-peel tester. The first, "Static Peak force" is generated as movement of the eggshell is initiated, i.e., it is the force required to release the grip of the tube of glue-like fixative that holds the eggshell in place on the hair (Fig. 2). Subsequently, as the glue tube slides along the hair some friction is generated that is averaged out by the device resulting in an "Average force" measurement (Fig. 2). Previous studies showed that the average force could be reduced by simple lubricants but Static Peak force was not significantly affected by any chemical treatments (*Burgess, 2010*; *Lapeere et al., 2014*; *Ortega-Insaurralde et al., 2014*).

## Formulation development

For the extensive primary screening process involving several hundred chemical substances, formulation components, mixtures, and commercial products, each candidate compound or mixture was tested for its effect on the Peak Force measurement using at least 10 louse eggs on hair for each replicate test. In each case the candidate material was compared with a suitable negative (damp or dry hair) and/or positive (a hypothetically lubricant preparation such as conditioner) control treated group of louse eggs. For the primary screening tests only intact louse eggs were used to ensure consistent rigidity of the eggshells under test. The search for a compound to facilitate louse egg removal had the obvious starting point of investigating the activity of known lubricants likely to facilitate sliding of the tube of glue-like fixative along the hair shaft after the initial release. We investigated various existing oily

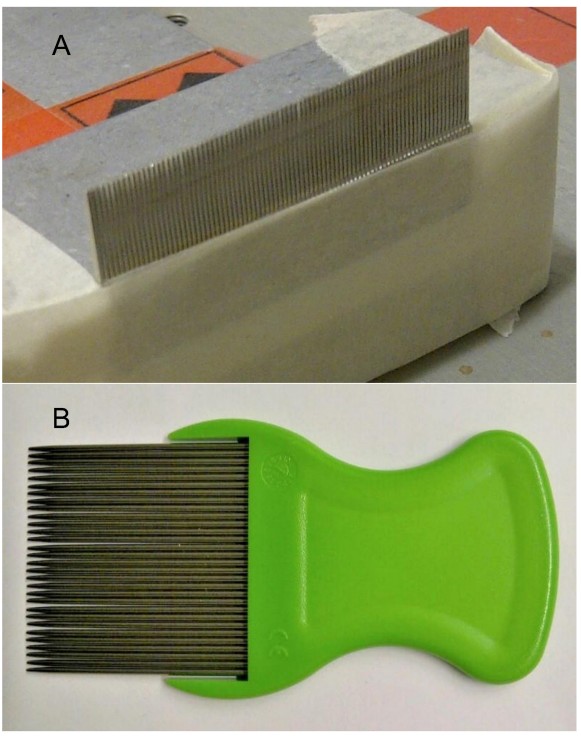

**Figure 1 Combs used in the study.** (A) Innomed™ Lice Comb attached to platen of the slip-peel tester. (B) The comb supplied in the Hedrin Stubborn Egg Remover Kit. Photo credit: Ian F. Burgess.

and surfactant-based cosmetic and toiletry products and head louse treatments, together with hair conditioning formulations and compounds. For unformulated chemicals, such as the oily and film-forming compounds, testing was conducted using undiluted material in the first instance, with subsequent dilutions in appropriate vehicles.

Each compound investigated was applied by immersing louse eggs on hair in the fluid for 30 min after which the eggs were subjected to the slip-peel test. The effects of the chemicals were evaluated both with and without a wash off procedure. Materials compatible with water were simply rinsed off but oily materials required a shampoo wash. The initial measure of effectiveness was whether a compound or mixture produced a lower Peak Force measurement than was observed with a simple hair conditioner (*Burgess, 2010*; *Lapeere et al., 2014*; *Ortega-Insaurralde et al., 2014*). Each material was initially tested using just one batch of 10 louse eggs. If it showed a consistency of effect, i.e., all the readings were similar without any extremely high Peak Force outliers, further batches were tested. In some cases, e.g., dipentaerythrityl pentaisononanoate (Fig. 3) and isononyl isononanoate + PEG-8 dimeticone phosphate (Fig. 4), there was considerable variability in effect so these chemical entities were rejected and no further batches were tested. Other examples, e.g., octyl palmitate and isopropyl myristate, were less clear, because some Peak force readings were quite low but other eggshells remained fixed to the hair so strongly that the hair broke before the eggshell started to move. For materials with these characteristics no more than two batches of louse eggs were investigated. All "formulated" mixtures, i.e., where

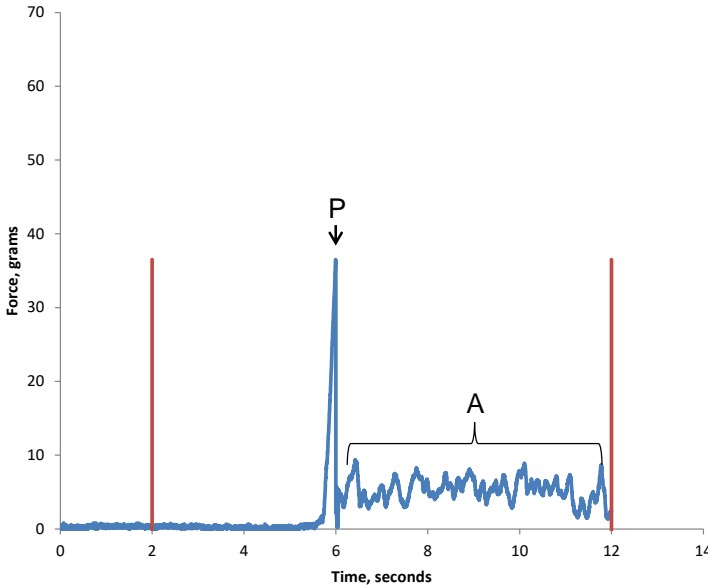

**Figure 2 Slip-peel tester output to show Static Peak force and Average force generated when removing a single louse eggshell.** Red lines show the start and stopping of the platen movement. Force output from the device is given in grams. The Static Peak force output (P) for initiating sliding of this egg was 348.1 mN and appears as a single peak. The average force (A) ranges between 16.3 mN and 81.7 mN and its relative lack of smoothness indicates some friction between the egg fixative tube and the hair.

potential active substances were incorporated into either existing preparations like 4% dimeticone gel or into new mixtures such as shown in Figs. 4 and 5, between two and five batches of eggs were used according to the consistency or variability of effect on the Peak Force measurement.

The final formulation, subsequently commercialized as a consumer product (Hedrin Stubborn Egg Loosening Lotion, Thornton & Ross Ltd., Huddersfield, UK) (*Cooper & Brunton, 2017*), and some variations on the relative proportion of the component chemicals of the formulation, was tested repeatedly, at least five times each, in order to confirm comparative effectiveness using separate batches of intact louse eggs as well as batches of hatched eggs (nits). The commercial formulation was also tested on one occasion in comparison with some marketed products from Europe and North America that claim to have egg removal capability or lubricant characteristics that make egg removal easier (see 'Results' for details of the products).

## Usage evaluation

Using the previously validated method of measuring egg and nit removal from single hairs by means of the slip-peel tester, it was relatively straightforward to demonstrate the effectiveness of the final formulation in the laboratory. However, this could not confirm effectiveness when used on a human head where numerous hairs would be combed through at the same time and where differences in hair density, thickness of the hair shafts, and other physical characteristics would have an uncontrolled effect on the presentation of the louse eggshells and the fixative to both the egg removal formulation

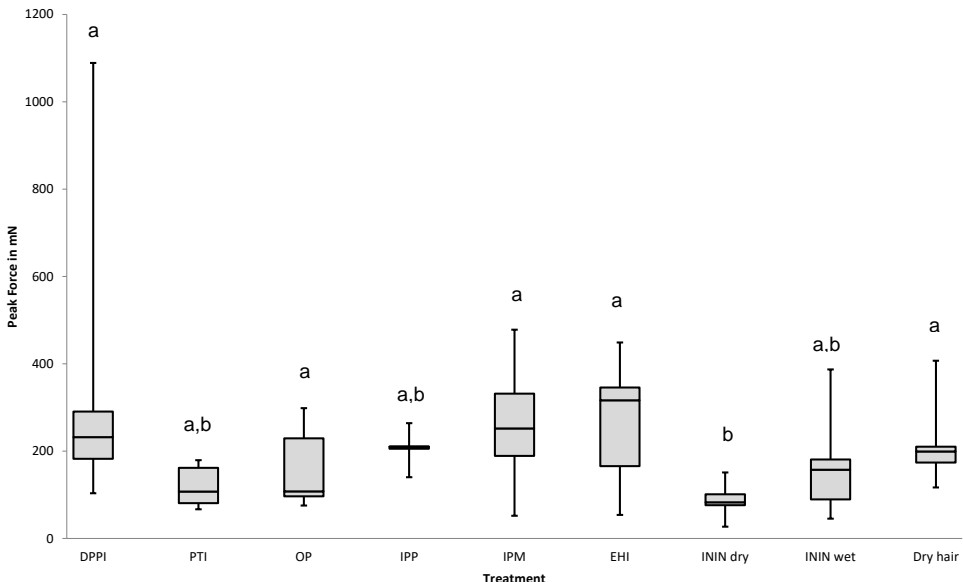

**Figure 3** **Effect of various alkyl esters on the peak force required to move louse eggs on hairs.** Key to treatments: DPPI, dipentaerythrityl pentaisononanoate; PTI, pentaerythrityl tetraisostearate; OP, octyl palmitate; IPP, isopropyl palmitate; IPM, isopropyl myristate; EHI, ethylhexyl isononanoate; ININ dry, isononyl isononanoate on dry hair; ININ wet, isononyl isononanoate on pre-wetted hair; Dry hair, dry untreated hair. Treatments with the same letter have no significant difference $p < 0.05$.

and the teeth of a comb. In order to demonstrate that the formulation could facilitate eggshell removal an untreated control was necessary. For this a half-head approach was selected on the principle that each participant would act as their own control, although it was recognized that there could be differences in the number and distribution of louse eggshells on one side of the head from the other. Participants in this study were recruited from prior contacts who had taken part in clinical studies and from respondents to radio advertising. For a study of a commercially available cosmetic or Class 1 medical device product, used for its intended purpose, there is no requirement or procedure in the United Kingdom to seek ethical approval through the National Research Ethics Service (see http://www.hra-decisiontools.org.uk/ethics/). However, the protocol employed was based on a protocol previously submitted by us to Huntingdon Local Research Ethics Committee (07/Q0104/44) for a similar procedure (*Burgess et al., 2017*) and was internally reviewed by the sponsor prior to being commissioned. The actual methodology of combing followed that used in a previously published study conducted elsewhere (*Gallardo et al., 2013*).

Prospective participants were provided with an information booklet describing the purpose and procedures of the study. Inclusion age was 4 years or over and, on an initial visual screening, to have at least 20 louse eggs and nits present. The age limit was set related to expected hair thickness characteristics rather than on safety grounds, because younger children often have relatively sparser hair. The only exclusions were: being pregnant or having a long term or irritant scalp condition, other than pediculosis. All participants satisfying the inclusion criteria either signed their own informed consent form, if over the

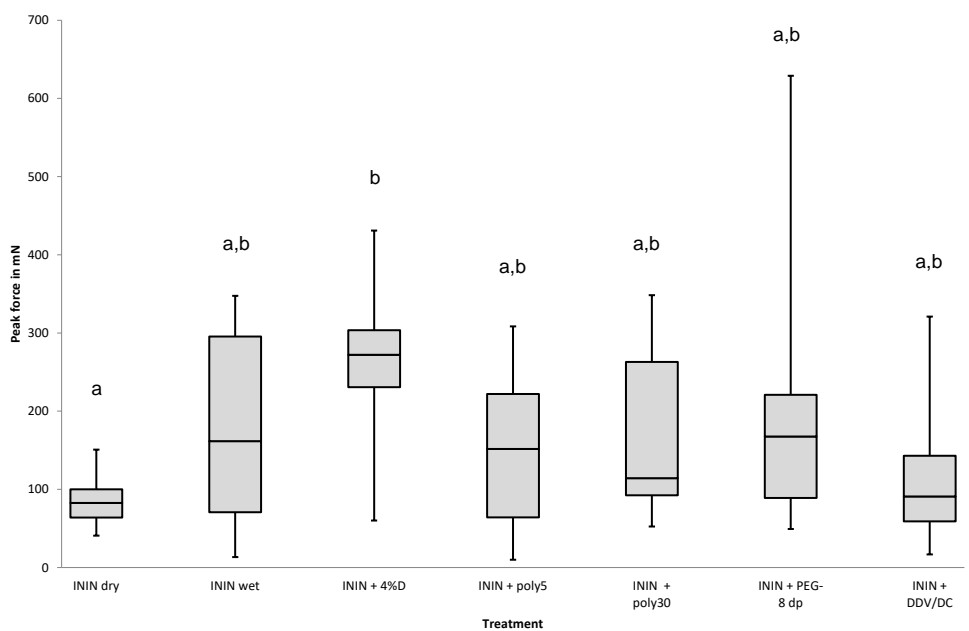

**Figure 4 Effect of various oily carriers on the activity of isononyl isononanoate to reduce the peak force required to move louse eggs on hairs.** Key to treatments: ININ dry, isononyl isononanoate applied to dry hair; ININ wet, isononyl isononanoate applied to pre-wetted hair; ININ + 4%D, isononyl isononanoate mixed into Hedrin Once 4% dimeticone spray gel applied to dry hair; ININ + poly5, isononyl isononanoate + polyisobutene 5; ININ + poly30, isononyl isononanoate + polyisobutene 30; ININ PEG-8 dp, isononyl isononanoate + PEG-8 dimeticone phosphate; ININ + DDV/DC, isononyl isononanoate + bis-divinyl dimeticone/PEG-10 dimeticone crosspolymer. Treatments with the same letter have no significant difference $p < 0.05$.

age of 16 years, or it was signed on their behalf by a parent/guardian and countersigned by the visiting investigator, additionally children under 16 gave written assent.

At the time of recruitment, potential participants were checked for the presence of live head lice. If low numbers of lice were present the person was not treated immediately so that the egg remover product could be evaluated without any possible interference from treatment product residues. In those cases the treatment was provided after completion of the egg remover test. Alternatively, especially where a large number of lice were present, the person was treated to eliminate infestation one week before the test of the egg remover, which allowed sufficient time for any residues of silicone to be washed from the hair.

After providing consent the participant washed their hair or it was washed by the parent/caregiver using a basic, non-conditioning shampoo, rinsed, and towel dried. Hair characteristics, e.g., thickness/fineness, dryness/greasiness, etc., were assessed prior to washing using a subjective assessment based on prior experience combing numerous children during previous head louse treatment studies and the texture/feel when combing with a grooming comb. All participants received the same treatment; although the side of the head treated using the egg removing product was determined using a computer generated randomization sequence. Allocation was blinded until each participant was
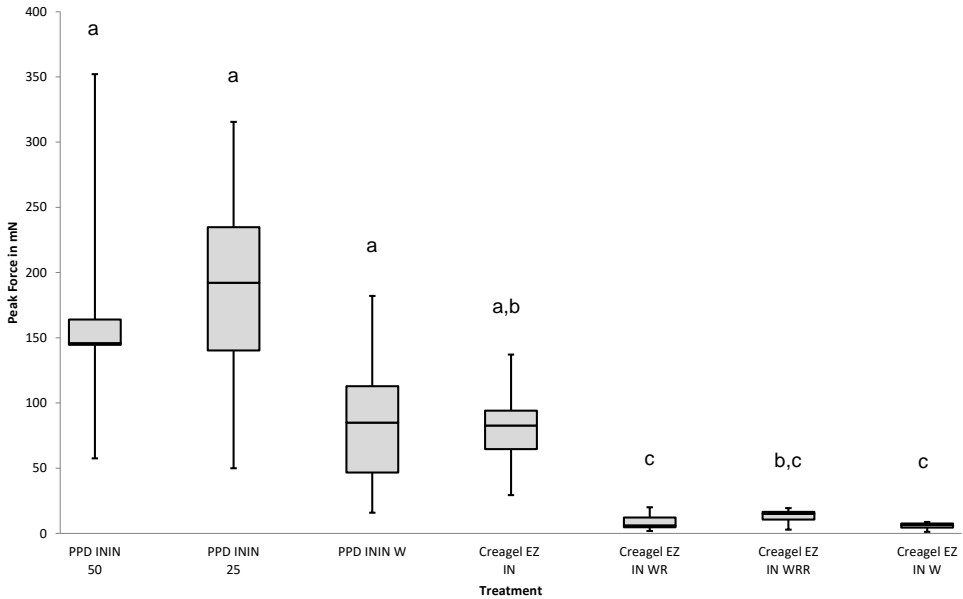

**Figure 5** Effect of PEG/PPG dimeticones and emulsified gel carriers on the activity of isononyl isononanoate to reduce the peak force required to move louse eggs on hairs. Key: PPD ININ 50, PEG/PPG-20/15 dimeticone + isononyl isononanoate in a 50:50 ratio; PPD ININ 25, PEG/PPG-20/15 dimeticone + isononyl isononanoate in a 75:25 ratio; PPD ININ W, PEG/PPD-20/15 dimeticone + isononyl isononanoate + water in a 60:15:25 ratio; Creagel EZ IN, Creagel EZ IN auto emulsified without added isononly isononanoate; Creagel EZ IN WR, Creagel EZ IN with added isononyl isononanoate + water in a 20:80 ratio followed by a water rinse; Creagel EZ IN WRR, Creagel EZ IN with added isononyl isononanoate + water in a 20:80 ratio applied to pre-wetted hair and then followed by a water rinse; Creagel EZ IN W, Creagel EZ IN with added isononyl isononanoate + water in a 20:80 ratio—no rinse. Treatments with the same letter have no significant difference $p < 0.05$.

randomized, the allocation being enclosed in a sealed, numbered instruction slip indicating which side should be treated.

The damp hair was parted along a center line and one half was then combed from scalp to tip using the comb (Fig. 1B) supplied as a package enclosure with the test product (Hedrin Stubborn Egg Remover Kit, Thornton & Ross Ltd, Huddersfield, UK) and after each stroke of the comb any louse eggshells removed were transferred to a medical wipe tissue. The comb, which is a registered Class 1 medical device in the European Union, was drawn systematically through that section of hair to remove as many eggs as possible using 10 strokes of the comb following a sequence essentially similar to that used by *Gallardo et al. (2013)*, i.e., four strokes from front to back, three strokes from back to front, and three strokes from above the ear to the crown of the scalp. The hair on the other side of the head was then treated using the egg loosening lotion, which was thoroughly spread through the hair using a comb with wide spaced teeth and left *in situ* for a timed 10 min before starting nit-combing. The hair on that section was then combed from scalp to tip using same sequence as on the untreated side. All louse eggs, nits, and glue fragments that were removed and extracted from the teeth of the comb were recovered to the case record and counted using a stereomicroscope.

## Analyses

Prior observations (*Burgess, 2010*; *Burgess, Brunton & Burgess, 2016*) showed unpredictable variations in peak force required to initiate movement of eggs or nits along hairs and that the data obtained using the slip-peel test method are not always normally distributed. All comparisons of the effect of different chemicals, chemical mixtures, or formulated materials on peak force measurements ex vivo were therefore considered non-parametric. The relative effect of treatments to facilitate reduction of peak force was analyzed using the Kruskal-Wallis rank sum test for multiple independent samples with post-hoc analysis by the Dunn method, adjusted by the false discovery rate (FDR) procedure of *Benjamini & Hochberg (1995)*. These tests were conducted using the calculator available online at http://astatsa.com/KruskalWallisTest/.

In the half-head study of louse eggs and nits removed from treated and untreated hair, the two halves of each head were not considered to be the same due to differences in hair styling and possible disturbance by the individual that could have resulted in fewer eggs being laid on one side. These paired comparisons were analyzed using a Wilcoxon Signed-Rank test. Separate analyses were performed for louse eggs, nits, and total numbers of eggshells using the calculator available online at https://www.socscistatistics.com.

# RESULTS

## Product development

Although initially conditioner-like materials, such as the medium-long chain alkanols hexadecane-1-ol or octadecan-1-ol, were considered possible facilitators of louse egg release, it was not possible to improve on the activity previously observed. Similarly, the supposed lubricant effects of siloxane (silicone) oils, such as found in head louse treatment products, did not alter the initial force required to start louse eggs moving along hairs confirming previous observations (*Burgess, Brunton & Burgess, 2016*; *Burgess et al., 2017*). Some other "oily" materials were found to reduce the mean Peak Force measured over a batch of louse eggshells, particularly some alkyl esters and emollient surfactants, although there was considerable variation of effect between compounds and some showed more variance of the Peak Force within a batch of eggs than when removing eggs from dry hair, examples of which are shown in Fig. 3. In addition, on three occasions when evaluating the effect of isopropyl myristate and twice when testing octyl palmitate, the louse egg under test remained firmly fixed to the hair so that the hair broke and the Peak Force readings, which were off the scale, were not included in the data shown. From this screen it was observed that the most effective compound for reduction of Peak Force was a derivative of pelargonic (nonanoic) acid, isononyl isononanoate, which appeared to reduce the Peak Force most and most consistently when applied to eggs on dry hair and reduced the Peak Force significantly ($p = 0.031432$) compared with dipentaerythrityl pentaisononanoate, isopropyl myristate, ethylhexyl isononanoate, as well as when removing eggs from dry hair. However, when the hair was pre-wetted, as would likely be the case when a caregiver chooses to remove nits from a child, it was less effective to reduce Peak Force and showed greater variance of effect (Fig. 3). We found incorporation of ININ into other oily materials

considered to have lubricant properties; such as polyethylene glycol (PEG) dimeticones, the head louse treatment 4.0% dimeticone gel, which also contains a PEGylated dimeticone, or siloxane alternatives such as polyisobutene compounds; actually reduced the activity of the ININ even on dry hair. The only significant difference in terms of Peak force across this group of mixtures was between ININ on its own, applied to dry hair, and ININ mixed into 4% dimeticone gel ($p = 0.035624$). All other comparisons were not statistically significant (Fig. 4).

Any product designed to facilitate louse egg removal should be applied to the hair and remain in contact for whatever time is required for the active principle to take effect. Because isononyl isononanoate (ININ) is oily in nature and water immiscible it is normally formulated with other compounds in order to make a cosmetically manageable preparation. A gel-like preparation was considered the most suitable. Initially a water soluble, PEG-modified silicone fluid, compatible with ININ, and designed for use in skin and hair products to provide a rinse off light conditioner was investigated. Although easy to wash off, in undiluted format the gel was sticky and relatively ineffective whether using a 50:50 or 75:25 mixture with ININ but when diluted to give a final mix of 60:15:25 egg removal improved (Fig. 5) as did lubricity, but the mixture was not physically stable. It was found that a number of cosmetic products contain polymeric gelling agents that are designed to be used for creating water in oil or oil in water emulsions are water miscible and auto emulsifying. This range of products sold under the brand name Creagel® (CIT S.a.r.l., Dreux, France) incorporates a sodium acrylate/sodium acryloyldimethyl taurate copolymer together with one or more of a hydrocarbon, ester, or fixed oil ready for auto emulsification with water. One product Creagel® EZ IN contains copolymer plus 10–30% ININ. Initial tests with only the emulsified polymer gave high peak force values, which suggested the ININ had a low bioavailability "locked" in the gel matrix, and this was not significantly improved even with inclusion of additional ININ. However, adding more water to the auto emulsified gel and then loading ININ to create a 20%–23% concentration in the final mixture was found to provide the best medium for delivery of the isononyl isononanoate and the mean Static Peak force was significantly reduced ($p < 0.022$) compared with the undiluted gel, enabling eggs and nits to be removed from the hair smoothly with minimal initial drag (Fig. 5).

The final formulation was developed as a commercial product (Hedrin Stubborn Egg Loosening Lotion, Thornton & Ross Ltd, Huddersfield, UK), which when tested in comparison with other products claimed to show efficacy against louse eggs was more effective to remove louse eggs with reduced forces (Fig. 5). For example, in the comparison with the group of marketed products from Europe and the USA claiming to aid egg and nit removal, the isononyl isononanoate gel was significantly more effective to reduce the Static Peak force required to initiate removal than K.O. Poux ($p = 0.000609$); Puressentiel Pouxdoux® ($p = 0.001416$); Nit Free Mousse ($p = 0.0017776$); Paranix Après Traitement and Ecrinal® Poux ($p = 0.008686$); but not significantly more effective than K.O. Lentes ($p = 0.060550$) or OTC Antipiojos ($p = 0.208917$). However, the mean peak force measurement for the isononyl isononanoate product (21.225 mN) was only one third

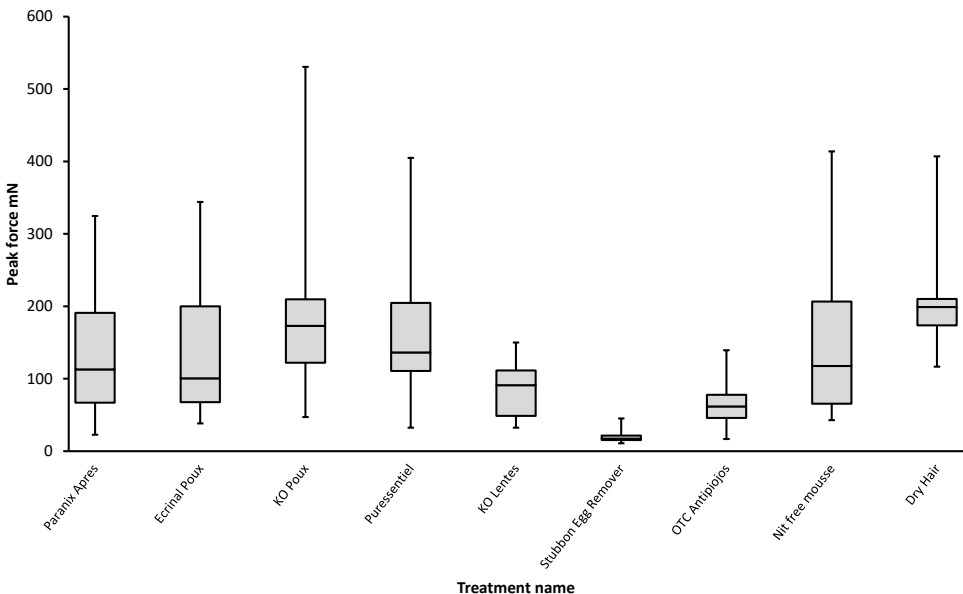

**Figure 6** **Static Peak force required to remove louse eggs after using various marketed products with indications for facilitating nit removal compared with Stubborn Egg Remover.** Products used were: Paranix Après Traitement, Omega Pharma, Nazareth, Belgium; Ecrinal® poux, Shampooing doux assainissant, Laboratoires Asepta, Monaco; K.O. Poux, Shampooing doux familial, Item Dermatologie, Nogent sur Marne, France; Puressentiel Pouxdoux®, Shampooing quotidien, Puressentiel France, Paris, France; K.O. Lentes, Baume décolleur et répulsif, Item Dermatologie, Nogent sur Marne, France; OTC Antipiojos, Acondicionador desprende liendres, Ferrer Internacional, S.A., Barcelona, Spain; Nit Free Lice and Nit Eliminating Mousse and Nit Glue Dissolver, Ginesis™ Natural Products, Waterloo, Alabama, USA.

of that of OTC Antipiojos (65.3625 mN) the most effective of the competitor products (Fig. 6).

## Usage evaluation

For the in vivo study, 15 participants were recruited from nine households (Table 1). All but two were aged 10 or over, mainly because this group had become more conscious of the presence of louse eggs and nits as being unsightly, particularly the girls who wished to wear their hair up from time to time, and only two were male.

When the Hedrin Stubborn Egg Loosening Lotion was tested in human volunteers using the comb supplied in the commercial treatment kit, the numbers of eggs and nits combed from each of the participants on wetted hair alone and after treatment with the egg remover product are shown in Table 2. In five cases, participants 003, 006, 008, 009, and 024, the number of recovered eggs and or nits was low because of irregularities in the alignment of comb teeth (Fig. 1B) that allowed some eggshells to pass between during combing. Each of these participants, apart from 003, was judged to have hair that is "Thick" (see Table 1), making it difficult to draw the combs through the hair because any inconsistencies in the alignment of the teeth snagged on the hair locks. It is possible that use of greater quantities of the egg removing product could minimize some of these problems but conducting the

**Table 1  Demographic characteristics of participants.**

| Household | Study number | Age | Sex | Hair characteristics | | | | | Side treated |
| | | | | Length | Thickness | Curl | Dryness | Colour | |
|---|---|---|---|---|---|---|---|---|---|
| H1 | 001 | 12 | M | ES | Medium | Straight | Normal | Brown | Left |
| | 004 | 16 | F | BS | Medium | Straight | Normal | Brown | Right |
| | 006 | 13 | M | AE | Thick | Straight | Greasy | Brown | Left |
| H2 | 002 | 13 | F | BS | Medium | Straight | Greasy | Brown | Left |
| | 003 | 10 | F | ES | Medium | Wavy | Normal | Blonde | Right |
| H3 | 007 | 11 | F | BS | Fine | Straight | Normal | Blonde | Right |
| H4 | 008 | 14 | F | BS | Thick | Curly | Normal | Brown | Right |
| | 009 | 13 | F | BS | Thick | Wavy | Normal | Black | Left |
| H5 | 011 | 12 | F | BS | Fine | Straight | – | Brown | Right |
| H6 | 014 | 7 | F | BS | Fine | Straight | Normal | Brown | Left |
| | 018 | 15 | F | ES | Medium | Straight | Normal | Brown | Left |
| | 022 | 18 | F | BS | Fine | Wavy | Greasy | Brown | Left |
| H7 | 015 | 17 | F | ES | Fine | Straight | Normal | Blonde | Right |
| H8 | 020 | 14 | F | BS | Fine | Straight | Normal | Brown | Right |
| H9 | 024 | 9 | F | BS | Thick | Straight | Normal | Blonde | Right |

**Notes.**

Sex: F, Female; M, Male.

Hair length: AE, Above Ears; ES, Ears to Shoulders; BS, Below Shoulders.

Hair thickness and dryness/greasiness were assessed subjectively based on prior experience.

study using excess gel would have made observation and recovery of the eggshells more difficult.

Eggs and or nits were removed from all participants from one or both sides of the head. Of the 15 participants, 14 (93.3%) had more eggs/nits or fixative fragments removed from the side of the head treated using the Stubborn Egg Remover. The one anomalous participant (015) had a large number of fragments of very old, brittle eggshells and nits that had undergone exposure to numerous treatments for infestation as well as other products applied to the hair. Consequently, the fixative material, which held these eggs and nits to the hair, was also very brittle and simply disintegrated when combed on the untreated side. Along with the broken eggs and nits a large number of fragments of the fixative also came away from the hairs and stuck to the comb. On the treated side, the egg removing gel made no apparent difference to the brittleness of the eggshells or the fixative. These also broke away from the hairs during combing but, because of the viscosity of the gel and irregular spacing of teeth on the comb, many fragments became trapped in the gel film so that most slipped through the spaces between the teeth of the comb and were lost in amongst the hairs.

Regardless of the difficulties encountered with some participants, the egg removing product, as well as acting as a lubricant to help slide the comb through the hair, was confirmed in its ability to make removal of louse eggshells easier and more straightforward. Comparison of the numbers of eggs and nits from treated hair with the numbers from untreated hair (Table 2) found an overall significant advantage ($p = 0.01046$) for the egg

**Table 2  Numbers of egg/nits combed from participants' hair.**

| Participant | Number of egg/nits removed during combing | | | |
| --- | --- | --- | --- | --- |
| | No treatment | | Stubborn egg remover | |
| | Eggs | Nits | Eggs | Nits |
| 001 | 35 | 32 | 87 | 49 |
| 002 | 2 | 0 | 13 | 5 |
| 003 | 1 | 1 | 6 | 1 |
| 004 | 0 | 1 | 27 | 38 |
| 006 | 0 | 0 | 1 | 0 |
| 007 | 6 | 2 | 68 | 47 |
| 008 | 0 | 0 | 0 | 1 |
| 009 | 1 | 1 | 1 | 3 |
| 011 | 0 | 0 | 16 | 9 |
| 014 | 0 | 0 | 8 | 3 |
| 015 | 59 | 127 | 4 | 7 |
| 018 | 0 | 0 | 8 | 3 |
| 020 | 6 | 2 | 34 | 11 |
| 022 | 4 | 0 | 8 | 2 |
| 024 | 0 | 0 | 4 | 0 |
| Totals | 114 | 166 | 285 | 179 |

loosener as well as significant advantages for removing intact louse eggs ($p = 0.01928$) and nits ($p = 0.034$).

## DISCUSSION

This is the first report of a product designed to remove the eggs of head lice that has demonstrated loosening of the grip of the fixative holding the eggshells to the hair shaft. Other studies have shown that some products can lubricate the hair to reduce or minimize snagging but in all cases the initial movement required some measure of force causing a tugging effect on the hair and scalp (*Burgess, 2010*; *Lapeere et al., 2014*; *Ortega-Insaurralde et al., 2014*). This study ignored preconceptions and took an empirical approach to the problem of identifying an egg loosening compound. Previous investigations showed that lubricants facilitated the sliding component of egg removal so this investigation centered on identifying one or more chemicals that could extend this effect into the initial loosening process. The discovery that pelargonic acid derivatives have an effect on the grip of the louse egg fixative was serendipitous. During the screening process several compounds in this group were shown to have some activity but isononyl isononanoate was identified as the most effective. In cosmetic products pelargonic acid derivatives are considered extremely safe and are used at concentrations from 0.03% to 64% (*Johnson Jr et al., 2011*), requiring further formulation appropriate to the application, in this case by jellification to provide a stable preparation that remains in full contact with the eggshells on hairs prior to nit combing. Polymeric gelling agents formed by hydration were found to be compatible carriers of these lipid-like materials, improving the flow characteristics and

increasing lubricity. Such emulgels are widely used in the cosmetic and pharmaceutical industries (*Samala & Sridevi, 2016*) and are particularly used for delivery of topically applied hydrophobic drugs in a cosmetically acceptable manner (*Ajazuddin et al., 2013*). The result was a gel that not only lubricated the sliding of the eggshells but more importantly was found to minimize the force required to initiate moving of the tube of eggshell fixative along the hair, something no previous formulation has achieved successfully.

Despite the fact that the majority of eggshells on a head are usually either hatched or non-viable, and therefore unlikely to result in continued infestation or reinfestation (*Williams et al., 2001*), many healthcare professionals, school officials, and parent/caregivers wish to see all eggshells removed to eliminate any doubt (*Altschuler & Kenney, 1986*). However, the problem with enforcing policies for removal of louse eggs, for example the "No Nit" policies operated by many North American school boards since the late 1980s, is that they can become draconian. These affect children, through unnecessary treatments and missing school, and parents through missing work and loss of earnings (*Williams et al., 2001*; *Mumcuoglu et al., 2006*). However, for many families in most countries the real problem with persisting eggshells is the fact that they are unsightly, revealing past infestations and suggesting that insufficient care has been taken in hair management. This also results in stigmatization and makes children, particularly girls, self-conscious and unwilling to style their hair in a way that could reveal the eggshells and nits.

There are numerous combs marketed for nit removal but many of them are poorly designed so that they are ineffective, uncomfortable in use, and in some cases cause damage to the hair shafts so they split or knot up (*Burgess, Brunton & Burgess, 2016*). As a result nit removal is considered an uncomfortable experience for both the receiver and the giver of the procedure. In several countries this inconvenience has reached the point where a proportion of consumers prefer to visit a specialist salon for louse treatment and nit removal rather than attempting to do this at home. However, such services come at a financial cost that may not be sustainable if a household experiences regular infestations.

Relatively cheap and effective treatment products with a physical mode of action have been available in most European countries, Israel, and Australia for some time. These eliminate lice when used correctly but do nothing to facilitate egg and nit removal (*Burgess, Brunton & Burgess, 2016*). Consequently, the discovery of a compound that releases the grip of the glue-like fixative holding louse eggshells to hair offers a new option to improve the effectiveness of all treatments by allowing easier removal of any eggs that may have been missed during a treatment. This should give parents and schools greater confidence in the possibilities for elimination of infestation as well as improving the wellbeing of the children by minimizing the requirement for repeated combing. For those people who prefer to treat a head louse infestation by combing methods, such as wet combing with conditioner, nit removing can also be stressful, so inclusion of a lubricating lotion into the procedure that also removes eggshells potentially makes this a one-stage treatment process because the loosened eggshells can also be removed by the plastic detection combs used for wet combing.

Use of this type of product will of course vary from one country to another depending upon the perceived cultural and social impact and necessity of removing the eggshells

after elimination of infestation. However, our experience of conducting clinical studies in the British community over a period of approximately 20 years has been that a growing proportion of families have become more conscious of persistent louse eggs and nits and increasingly want to remove them more efficiently. Having a treatment that makes egg and nit removal easier may have some small impact on levels of infestation overall but its main function will be empowering those families that have hitherto largely given up on attempts to remove louse eggshells for cosmetic reasons as being too great a challenge to manage. The result should be an improvement in self-esteem for those girls currently too embarrassed to style their hair, or in some cases too embarrassed to attend school, because they know that others are able to see the old eggshells stuck to the hairs.

## CONCLUSIONS

Removal of head louse eggshells from hair is hampered by the grip of the egg fixative on the hair shafts. It has been shown previously that many products claiming to release the eggs and nits from hair do not work and there is a need for a preparation that facilitates eggshell removal. This study found a select group of compounds that loosened the grip of the fixative and, when formulated into an appropriate aqueous gel, could be shown in the laboratory to reduce the force required to initiate movement of the eggshells along a hair. It was also shown to enable easier removal of eggs and nits in a small usage evaluation in comparison with combing hair wetted only with water. In those territories where "nit removal" is considered an integral part of treatment such a treatment should make the process less stressful and easier for most households.

## ACKNOWLEDGEMENTS

Our thanks go to all the chemical supply and manufacturing companies who kindly gave us samples to test in this project. Also we would like to thank the individuals and families who participated in the combing study. This manuscript has been revised and corrected thanks to the comments from three reviewers.

### Funding

This work of research and development was self funded by EctoMedica Limited. Funding for the preparation of the publication and publication fees was provided by Thornton & Ross Ltd. The funders had no role in study design, data collection and analysis, decision to publish, or preparation of the manuscript.

### Grant Disclosures

The following grant information was disclosed by the authors:
EctoMedica Limited.
Thornton & Ross Ltd.

## Competing Interests

Elizabeth Brunton, Ian Whelan, and Ian Burgess are shareholders in EctoMedica Limited. Elizabeth Brunton and Ian Burgess are directors of EctoMedica Limited. Ian Whelan is a director of Avisius Research Limited. Since this work was conducted, EctoMedica Limited has sold world rights to this formulation to Thornton & Ross Ltd. Insect Research & Development Limited provides paid consultancy services to Thornton & Ross Ltd.

## Author Contributions

- Elizabeth R. Brunton conceived and designed the experiments, performed the experiments, analyzed the data, authored or reviewed drafts of the paper, approved the final draft, investigated all potential active materials.
- Ian P. Whelan conceived and designed the experiments, performed the experiments, contributed reagents/materials/analysis tools, approved the final draft, researched potential excipient media.
- Rebecca French performed the experiments, organised clinical investigation.
- Mark N. Burgess performed the experiments, contributed reagents/materials/analysis tools, evaluated potential active materials.
- Ian F. Burgess conceived and designed the experiments, performed the experiments, analyzed the data, prepared figures and/or tables, authored or reviewed drafts of the paper, approved the final draft.

## Human Ethics

The following information was supplied relating to ethical approvals (i.e., approving body and any reference numbers):

For a study of a commercially available cosmetic product or a CE marked medical device (this product is a medical device in some EU countries), used for its intended purpose, there is no requirement or procedure in the United Kingdom to seek ethical approval through the NHS ethics system. The protocol employed was based on a protocol previously submitted to Huntingdon Local Research Ethics Committee (application number 07/Q0104/44) for a similar procedure.

## Patent Disclosures

The following patent dependencies were disclosed by the authors:

Acevedo KBV. Formula for removing lice and nits and process for manufacturing the same. Mexican Patent application, MX 2010010985 A, 2010-10-06.

Bernstein JE. Method and composition for treating pediculosis capitis. United States Patent, US 4,927,813, 1990-05-22.

Cooper N, Brunton ER. A louse ova and nit loosening composition. UK Patent application GB2550558(A), 2017-11-29 https://patents.google.com/patent/GB2550558A/en.

Hayward JA, Watkins DC. Composition containing protease separate from glycosidase for removing nits in treating lice infestation. United States Patent, US 5,935,572, 1999-08-10.

Kolender E, Kolender B. Composition for treatment and prevention of lice. United States Patent application, US 2017/0049110 A1, 2017-02-23.

McGuire TM, Kross RD. Lice remover composition. United States Patent application, US 2002/0025336 A1, 2002-02-28.

Mehlhorn H, Schmahl G, Schmidt J, Abdel Ghaffar F, Al Rasheid K, Quraishi S, Al-Farhan A. Compositions comprising flavonoid-containing extracts from plants of the genus *Citrus* and/or isolated citrus flavonoids and specific cationic surface active agents, and said composition for use as an agent for treating infestations with head lice. United States Patent application, US 2013/0252911 A1, 2013-09-26.

Ozelkan S, Zhang Z, Malayev V. Method, composition and kit to remove lice ova from the hair. United States Patent, US 6,524,604 B1, 2003-02-25.

Sacker J. Improvements in or relating to hygienic combs made of metal. Great Britain Patent 550,636, 1942-04-09.

Upton HF. Method for removing nits from hair. International Patent WO 94/23690, 1994-10-27.

## Data Availability

The raw data are available in Tables 1 and 2.

## Supplemental Information

Supplemental information for this article can be found online at http://dx.doi.org/10.7717/peerj.6759#supplemental-information.

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
