# Peer review of "Head louse egg and nit remover—a modern “Quest for the Holy Grail”"

_PeerJ, doi:10.7717/peerj.6759_

## Round 0.1 · original submission · Major Revisions

My apologies for the delay, but it was not easy finding available reviewers. I received two complete reviews and the comments of a third expert, who indicated:

"...It is critical that researchers name the combs that they use for their work because their effectiveness varies greatly (see… http://ww1.prweb.com/prfiles/2018/02/10/15142040/NPA%20Petition%20to%20FDA%20sent.pdf ) and differentiate between what is actually effectiveness from combing vs chemical release of the nit glue.

Removal of nits is far more than a "cosmetic issue" and the statement shows an immediate bias about nit removal relevance in controlling head lice and protecting against overtreatment with pesticide treatments. Cosmetics also do not require FDA approval to the best of my knowledge and framing nit removal in cosmetic terms may help circumvent FDA scrutiny.

15 subjects is a weak sample for such a tall claim. Who did the combing for the data? What was the protocol for combing? The work needs to be reproduced with a larger amount of participants with work done by independent researchers."

As you can see, the opinion of the three experts is contradictory in some aspects, but all of them underline the fact that further information is needed.

·

Basic reporting

Head lice are specific and permanent parasite insects that live in the human head. Their eggs are firmly attached to the hair by means of a resistant glue-like substance, that holds the empty eggshell fixed to the hair long after the nymph has emerged.

In this article, authors evaluated in a laboratory test, the effectiveness of a formulation based on isononyl isononanoate to facilitate egg and nit removal from the hair, in a comparative study with various commercial products.

Moreover they evaluated in an in vivo test, the number of eggs and nits removed by combing for the head of 15 children in which one half of the head was treated with the formulation and the other half was combed only on damp hair.

Experimental design

The manuscript is relevant due to the cosmopolite medical importance of pediculosis and falls within the scope of the journal.

It is clear and well written, the introduction is adequate to the objectives of the article, and the bibliography is updated.

The methodologies are based on previous reports and are carefully described with sufficient information in the article.

The results are summarized in two figures and two tables, all of them are relevant to show the obtained results and the statistical analysis performed.

The conclusions are consistent with the results and respond to the objectives of the study.

Validity of the findings

However, I consider that some information is missing in the table 2 where the demographic characteristics of the participants are described.

Although some of the characteristics of the children are simple to determine (sex, age), and others are well described (length, colour and curl of the hair), there are others that are not well defined.
For example, it is not clear how authors established the thickness and the dryness of the hair.

Y strongly recommend to the authors to add some indication or criterion to define both variables in Table 2. This information will be useful in the application of this procedure in new studies.

Reviewer 2 ·

Basic reporting

The article is written in correct English. However, the manuscript is not clear in many paragraphs, mainly in the Materials and Methods section (see General Comments for the author).

The manuscript include sufficient introduction and background and is structured in a standard way. Relevant prior literature is appropriately referenced.

The figures and tables presented are consistent with the correspondent studies, however, there are studies/evaluations carried out that are not presented in tables, figures or text; that is, all the results that correspond to all the bioassays described in Mat and Met are not reported in the manuscript
(see General Comments for the author)

Experimental design

The identification and description of the knowledge gap is well presented, and this gap is relevant and meaningful; however, the objective of the study is not clearly established in the introduction. I suggest that the authors present, at the end of the introduction, clearly and concisely the objective of the study and, if possible, the hypothesis to be corroborated.

There is no clarity in the description of the experimental design and bioassays of the laboratory studies. I suggest that the authors: a) improve the order in which the experimental design and bioassays are described, b) reduce/unify paragraphs where the experimental design and the laboratory bioassay are described (a large number of paragraphs, where some descriptions are repeated, conspire against the expository clarity).
(see General Comments for the author)

Validity of the findings

The data and results of a large number of bioassays described in Materials and Methods (e.g. the primary screening and the evaluation of different versions of the formulation chosen to obtain an optimal final formulation) are not presented in the manuscript in any formal way (i.e. text, figures or tables). These results are only referred to in sentences without any quantitative and statistical information (see General Comments for the author).

On the other hand, there is no information on the statistical tests used, except in one case (lines 264-272) (see General Comments for the author)

The authors make an excellent discussion of the problem of the persistence of lice eggs in the hairs, even if the embryo has already died. However, they do not discuss their own experimental results, neither in terms of the meaning of themselves nor in relation to results of other studies (except lines 277-280). Nor do they discuss whether their results corroborate or refute the hypotheses proposed. Finally, the authors also do not attempt to discuss how the original hypothesis, or some alternative hypothesis that has arisen from unexpected results, allow explaining and/or can be explained by other results and hypotheses in their study area.

Additional comments

COMMENTS OF THE MANUSCRIPT: HEAD LOUSE EGG AND NIT REMOVER – A MODERN “QUEST FOR THE HOLY GRAIL” by Elizabeth R Brunton, Ian P Whelan, Rebecca French, Mark N Burgess, Ian F Burgess

This article presents laboratory studies of the effectiveness in the removal of eggs and nits from the head louse of different compounds and formulations. In a second part and after a selection based on the results of the initial screening, the article reports on the effectiveness, based on laboratory bioassay and in a kind of clinical study, of the selected compound and its final formulation.

The manuscript has serious problems in the presentation and dicussion of the study that prevent me from recommending it for publication. These problems can be summarized as follows (details are below in this review):

1) The objective of the study is not clearly established in the introduction.
2) There is no clarity in the exposition of the bioassays and experimental design of the laboratory studies.
3) There is no description of the statistical tests used.
4) No data, quantitative results and statistical tests of the primary screening, formulations and of the evaluation of the variations on the relative proportion of the components of the formulation are presented. In general, these results are only referred in sentences without any quantitative and statistical information. This is, perhaps, the most problematic of the manuscript. The only reported results in a formal way of the laboratory bioassays are the evaluations of the final formulation compared to some marketed products from Europe (i.e. Fig. 2).
5) Finally, the manuscript seems to be the presentation of a study that starts from the evaluation of a great variety of compounds (whose results are not shown in full), after which one candidate is selected and evaluated; but, after reading table 2 it is observed that the final candidate is a commercial product in the market; so the objective of the study is not clear.

Specific comments
Abstract
Line 26-30. This description is more appropriate for the results section
Introducion
Lines 57-59. Please, give some bibliographical reference.
Lines 67-68. I am not against the use of comparisons or other literary resources (i.e. rhetorical figures, tropos, etc.) but, since it is not recommended or optimal for a scientific manuscript, then it has to be used exceptionally; that is, in cases where the comparison or analogy used is adjusted in a very precise and unambiguous way to the study developed allowing a better understanding of the phenomenon studied, the question asked or the proposed objective (precisely, this requirement explains its infrequent use in manuscripts scientists and their unadvised use). In this case: a) the analogy between "nit remover" and "holy grail" is not entirely clear; b) the phrase "much myth and mystery as the original" is not very precise; e.g. How was the amount of myth and mystery quantified? In this way, I think that this comparison/analogy between the "nit problem" and the "mystery of the holy grail" does not make much sense since it does not help to better understand the proposal and the development of the research.
Materials and Methods
Lines 85. -18 ° C, does not affect the structure of the glue-like substance?
Lines 102-108. a) Writing should be improved to clarify the experimental design. For example, first a primary screening was carried out for evaluate the x1, x2, x3, and x4 compounds. Then, the selected compound was formulated with the y1, y2, y3 or y4 excipients and each formulation was evaluated. Different combinations and concentrations of excipients were also evaluated. Finally, the final formulation obtained was compared with commercial products z1, z2, z3, z4 and z5. The bioassay 1, which consisted of…., was used for the primary screening. The bioassay 2, which consisted of…., was used for the evaluation of final formulation.
b) What was the criterion to select the optimal compounds after primary screening?
Lines 118-120. Please, give some bibliographical reference.
Line 127. a) Why were only intact eggs used in this part of the study?.
b) Why do the authors give a bibliographic reference for the final formulation? Is this final formulation the result of the bioassays of the present research or is it the result of a study already published? Actually, this doubt is key in the understanding of the study.
Line 137-138. The phrase is not understood.
Lines 177-184 and 186-195. I suggest fusing both paragraphs.
Results
Lines 201-211. The authors do not present data, results and statistical information of each measured variable in each bioassay carried out; they only describe, without some numerical data, the obtained in a general way.
Lines 213-214. It is not an appropriate phrase for the results section, but rather it is a phrase for the introduction or for the discussion.
Lines 213-220. It is the first time that the authors refer to: 1) the vehicles that they used for the different formulations, 2) to the presence of a gel and 3) to that this gel was diluted with water. The authors should detail in the Mat and Met section the vehicles used to obtain each formulation, the different combinations they made, and the manner in which the selected formulation was diluted with water.
On the other hand, again there are no data, quantitative results and statistical significance of all the bioassays carried out.
Line 222-227. a) If Figure 2 is an example of the comparisons made, what were the other comparisons? What results were obtained?.
b) What does "p = 0.000764 to 0.006132" mean after saying that "Compound X was significantly more effective"?
c) In figure 2 it is not clear which is the compound that contains "isononyl isononanoate"
d) When reading table 2 it is clear that the product evaluated is "Stubborn Eggs remove"
1) Is this the "final formulation" referred to in the Mat and Met section?
2) Is this the formulation that contains isononyl isononanoate? If so, it should be specified in the Mat and met section.
3) Is this "final formulation" a commercial product? If so, it obviously does not seem to be an experimental formulation as described in the Mat and Met section.
4) And if the previous point is correct, then the study should be presented in another way
Lines 263-271. The statistical development is not clear (I insist that this must be described in the Mat and Met section). The design requires an analysis of paired comparisons. This analysis will be parametric or nonparametric according to whether the errors fulfill the assumptions of normality and homoscedasticity. The errors meet or not such assumptions, and there are objective evaluations based on specific tests. In this way, the decision to use parametric or non-parametric tests is not subjective but follows criteria established and shared by the scientific community. For all this, it is not correct to express "if we consider parametric, we have this result; if we consider it non-parametric, we have this other result. "
On the other hand, the Mann-W is not a nonparametric test for paired comparisons; one of the most used is Wilcoxon Test for paired comparisons

---

## Round 0.2 · Minor Revisions

Thank you very much for the improvements introduced in the manuscript. There remain, however, some minor points needing your attention concerning the presentation of the results, as indicated by the reviewer.

Reviewer 2 ·

Basic reporting

Regarding my previous revision:
The manuscript has improved notably, mainly the Materials and Methods and Results sections.
Minor revision are needed (see General Comments for the author)

Experimental design

Regarding my previous revision:
The objective is clearly established in the introduction.
The description of the experimental design and bioassays of the laboratory studies has improved markedly.

Validity of the findings

Regarding my previous revision:
The authors incorporated new information into the results that have improved the presentation and understanding of the study. Minor revisions are needed (see General Comments for the author)
The Discussion section was also improved

Additional comments

COMMENTS OF THE MANUSCRIPT: HEAD LOUSE EGG AND NIT REMOVER – A MODERN “QUEST FOR THE HOLY GRAIL” (#32808) by Elizabeth R Brunton, Ian P Whelan, Rebecca French, Mark N Burgess, Ian F Burgess
The revised manuscript has been significantly improved and is now suitable for publication in PeerJ. I thank the authors for their good work and for attending to all the comments and questions I have made..
Minor revisions are needed (but important when the development and effectiveness of a product is scientifically presented)
1) Please, the authors should clarify how many replicate (i.e. groups of 10 eggs) were made for each compounds/treatment in the bioassays of the figures 3, 4, 5 and 6.
2) Please, the authors should present the results of the tests with 5, 10 and 30 minutes of application described in lines 151-153. If these results are not presented, the authors should eliminate this description.
3) Lines 255-298. Please, the authors should refer to significant or non-significant differences between treatments by post-hoc comparisons of the bioassays corresponding to figures 3, 4 and 5 (the only statistical reference for these bioassays is on line 296). That is, the authors must specify the statistical significance (in words and/or with the P-values) of the results/comparisons described in these paragraphs.
4) Figs 3, 4, 5 and 6. The authors should show the significant differences between treatments in figures 3, 4, 5 and 6 (e.g. by means different letters).

---

## Round 0.3 · accepted · Accept

Thank you for improving the manuscript.